# Comparative Study on the Volatile Organic Compounds and Characteristic Flavor Fingerprints of Five Varieties of Walnut Oil in Northwest China Using Using Headspace Gas Chromatography-Ion Mobility Spectrometry

**DOI:** 10.3390/molecules28072949

**Published:** 2023-03-25

**Authors:** Lina Sun, Yanlong Qi, Meng Meng, Kuanbo Cui

**Affiliations:** 1Institute of Agricultural Mechanization, Xinjiang Academy of Agricultural Sciences, Urumqi 830000, China; 2Comprehensive Experimental Field of Xinjiang Academy of Agricultural Sciences, Urumqi 830000, China; 3College of Food Science and Engineering, Tianjin University of Science and Technology, Tianjin 300453, China

**Keywords:** walnut oil, volatile organic compounds, HS–GC–IMS, flavor fingerprint

## Abstract

Odor is an important characteristic of walnut oil; walnut oil aromas from different varieties smell differently. In order to compare the differences of volatile flavor characteristics in different varieties of walnut oil, the volatile organic compounds (VOCs) of walnut oil from five different walnut varieties in Northwest China were detected and analyzed using headspace gas chromatography–ion mobility spectrometry (HS–GC–IMS). The results showed that 41 VOCs in total were identified in walnut oil from five different varieties, including 14 aldehydes, 8 alcohols, 4 ketones, and 2 esters. Walnut oil (WO) extracted from the “Zha343” variety was most abundant in VOCs. The relative odor activity value (ROAV) analysis showed that aldehydes were the main aroma substances of walnut oil; specifically, hexanal, pentanal, and heptanal were the most abundant. Fingerprints and heat map analysis indicated that WO extracted from the “Xin2”, “185”, “Xin’guang”, and “Zha343” varieties, but not from the “Xinfeng” variety, had characteristic markers. The relative content differences of eight key VOCs in WO from five varieties can be directly compared by Kruskal–Wallis tests, among which the distribution four substances, hexanal (M), hexanal (D), pentanal (M), (E)-2-hexanal (M), presented extremely significant differences (P<0.01). According to the results of the principal component analysis (PCA), WO extracted from the “Zha343” variety was distinct from the other four varieties; in addition, WO extracted from the “Xin2” variety exhibited similarity to WO extracted from the “185” variety, and WO extracted from the “Xinfeng” variety showed similarity to WO extracted from the “Xin’guang” variety. These results reveal that there are certain differences in the VOCs extracted from five different WO varieties, making it feasible to distinguish different varieties of walnut oil or to rapidly detect walnut oil quality based on its volatile substances profile.

## 1. Introduction

Walnuts (*Juglans regia L.*), together with almonds, cashews, and hazelnuts, are the world’s four most popular tree nuts; walnut is one of the key economic forest trees in China. Walnut kernels are rich in oil (65–70%) and contain large amounts of polyunsaturated fatty acids (61.8–75.3%), including 62.2% linoleic acid and 13.5% oleic acid. Moreover, walnuts also contain a variety of vitamins and mineral elements [1,2]. Walnut ingredients are safe and suitable for long-term consumption [3,4]. Flavor is an important sensory characteristic of oil, which directly affects its acceptance by consumers. Contemporary research on the flavor of walnut oil has mainly focused on the composition of odor compounds [5,6]. However, there is few report on the differences in volatile components and key flavor substances among different varieties. So, it is going to be crucial to select a scientific analytical method for the extraction and comparative study of VOCs of WO.

Multiple advanced methods have been developed and used to analyze volatile organic compounds, such as but not limited to the following methods: (1) gas chromatography-mass spectrometry (GC–MS), which limits the extraction of high boiling point compounds and has a limited separation efficiency of one dimensional GC–MS when it comes to the trace compounds [7]; (2) comprehensive two-dimensional gas chromatography–mass spectrometry (GC × GC–MS) [8], which increases resolution and the significant separation improvement over 1D GC but provides large peak capacity although there is still some overlap [9,10]; (3) electronic nose [11], which is difficult to replace either complex analytical equipment or odor panels but supplements both of them, relying on the complex and diverse sensor array; additionally, its sensitivity and specificity are lower than MS. Among many VOC detection technologies, HS–GC–IMS has a distinct advantage in terms of the high resolution provided with gas chromatography and the high sensitivity conferred with ion transfer spectroscopy for the rapid detection of trace VOCs in samples [12], and no sample pretreatment [13,14]. Recently, with the development of an ion mobility spectrometry (IMS) library and the increase in relevant researchers, this technology has been greatly expanded and applied in the fields of food safety detection [15,16,17], environmental monitoring [18], medical diagnosis [19], and biomolecular determination [20,21,22]. Generally, GC–MS is recommended for the qualitative and quantitative analysis of VOCs, and GC–IMS is preferred to contrast differences between samples, due to its advantage of atlas visualization. Consequently, in this study, the VOCs of walnut oil samples from five of the most widely planted walnut varieties in Xinjiang were determined and analyzed using HS–GC–IMS. Further analyses were carried out using ROAV and PCA to identify and compare the key aroma components of walnut oil from different varieties. This study explored the feasibility of effectively distinguishing different varieties of walnut oil based on volatile substances and provided a theoretical basis for walnut oil production, processing, and quality control.

## 2. Results and Discussion

### 2.1. GC–IMS Spectrum Analysis

A top view of the three-dimensional spectrum generated using the reporter plug-in program in LAV analysis software is presented in Figure 1a. In order to perform a visual comparison of the sample spectrum, the spectrum of one sample was selected as the reference, which was then deducted from the spectrum of other samples [23]. For convenient observation, we chose the spectrum of sample A to be the reference. As shown in Figure 1b, if the VOC content is consistent with reference sample A, it is white after color deduction; red or blue indicate that the concentration of the substance is higher or lower than that in the reference, respectively. From Figure 1b, it can be observed that after color deduction, some differences in spectrograms of the five samples were reflected by the presence and depth of red and blue. The majority of characteristic signals of the five samples mainly occurred at the retention times of 100−300 s, this is similar to the analysis result of perilla seed oil [24]. Specifically, in the green square, the spectra of samples C and D showed more blue data points, whereas spectra of samples B and E showed more red data points, although the color of sample B was much lighter. In the red square, spectra colors of samples B, C, and D were nearly white, with only a few scattered red data points. In addition, compared with sample A, the VOCs marked with black and red circles were of lower content in sample B, C, D, and E, and the VOCs marked with pink and purple circles were of higher content. This was particularly notable for sample E.

### 2.2. Comparison of VOCs Fingerprints in Walnut Oil from Five Different Varieties

In order to clarify which substances were different, the characteristic flavor fingerprints of five different varieties walnut oil were constructed in the LAV program. In Figure 2, each row represents all signal peaks selected from a walnut oil sample, and each column represents the signal peak of the same VOC in different walnut oil samples. Letters M and D after the substance name represent monomers and dimers of the substance, respectively. All peaks were selected for comparative analysis. It can be seen directly from Figure 2 that the VOCs of walnut oil from five varieties were quite different. Specifically, substances in the red frame had the highest content in sample E. The content of 1-penten-3-ol in sample D was the highest. The content of substances in yellow frame was the highest in sample B. The content of substances in orange frame was the highest in sample A. However, there were also common VOCs among walnut oil from different varieties, such as hexanal (which was considered to be an important indicator of flavor deterioration, since its level was positively correlated with bitter and rancid taste [25]), pentanal (with the coffee, nut-like odor [7]), 1-octen-3-ol (which presented mushroom and earthy notes and could be used as the potential marker of walnut oil quality [7]), ethanol (which resulted from aerobic fermentation in the olives samples from Spain [17]), and undefined compound “9”. VOCs were the most abundant in walnut oil E. The diversity in these flavor substances makes it possible to distinguish between different varieties of oil.

### 2.3. Qualitative Analysis of VOCs of Walnut Oil from Five Different Varieties

As shown in Table 1, 50 VOCs in total were detected in five different varieties of walnut oil; 41 VOCs were identified, including 14 aldehydes, 8 alcohols, 4 ketones, and 2 esters. Hao et al. [7] reported that 50 volatile compounds were tentatively identified and quantified in walnut kernel samples using GC–MS, which included 14 aldehydes, 7 alcohols, 4 alkenes, 5 pyrazines, 3 furans, 1 acids, 5 esters, 5 ketones, and 6 other compounds. Among them, 12 compounds are consistent with the results of this study, except for the 9 unidentified components. Considering that his results were for the VOC detection of raw walnut kernels while this study is for the VOC detection of cold pressed walnut oil, it is possible that certain changes in the VOCs may occur during the processing of walnut oil. Certainly, further research is needed.

Moreover, it is also consistent with a prior report by Martỉnez [26] in which saturated and unsaturated aldehydes were most common VOCs in walnut oils, representing 55% to 81% of the total VOCs. The major aldehyde components present in virgin walnut oils are theoretically related with hydroperoxide precursors, which produce hexanal, 2-heptenal, octanal, nonanal, 2-decenal, and 2-undecenal [5,27].

The highest relative content among these VOCs was hexanal (relative contents: sample A, 34.36%; sample B, 33.40%; sample C, 42.30%; sample D, 43.30%; and sample E 38.52%), which is related to a green descriptor [28] and identified as an important aroma-active compound in WO [8].

### 2.4. PCA of Walnut Oil from Five Different Varieties

PCA is a dimensionality reduction method which determines the most contributing factors and the principal components. A spatial distribution map of the PCA can best reflect the differences between components [29]. The ionic peak intensity was taken as the characteristic parameter variable of PCA; it was performed on 15 samples from five different varieties (shown in Figure 3). The figure visualizes the differences between different samples. As shown, principal component 1 (PC1) (50%) and PC2 (33%) were extracted, adding up to 83%, which accounted for the vast majority of the original data information. The graph visualizes the differences between samples; a short distance between the samples means a small difference, and a long distance means a clear difference [24]. Furthermore, there was no overlap between samples, and all the data points showed regional distribution characteristics. This indicated that the VOCs of each sample were quite different. The data point distance between sample E and sample A was the largest; the distance between sample E and sample B was second largest; and the distance between sample E and sample C was the smallest. In addition, data points of sample E were distributed in the first quadrant; these data indicated that sample E was quite different from the other four samples. According to the data point distribution distances, sample A showed similarity to sample B, and sample C showed similarity to sample D.

### 2.5. Heat Map Analysis

In order to more accurately determine the differences in VOCs for different varieties of walnut oil, a heatmap was drawn using the R studio software (v1.2.13). This reflected the differences in the concentration of volatile compounds in the tested samples.

As shown in Figure 4, 50 VOCs were compared in the generation of a heatmap. Each row in the figure represents a different aroma compound, and each column represents a different oil sample. The dark red represents a high content; light red represents a low content; dark blue represents a low content; and light blue, as well as blanks, represent no detection. When a substance in the sample is exclusive or its content is higher than other samples, the substance can be designated as a characteristic marker of the sample. Considering the large amount of information and complex identification of all sample characteristic flavor substances in the fingerprint comparison, the differences in characteristic flavor substances in different varieties of walnut oil samples were further analyzed through heat map analysis to obtain their characteristic VOC markers. Figure 4 clearly indicates that, except for nine unidentified compounds, isopropyl acetate was the characteristic marker of sample A, which confers that fruity notes [7] belongs to the ester family. Most esters in WO are formed through esterification of various alcohols and carboxylic acids [8]. Acetone, 3-methylbutanol, and 1-hexanol (presented herbaceous and green aromas [7], which was also found in roasted almonds [30]) were the characteristic markers of sample B, and alcohols in WO are presumably produced by fat oxidation or aldehyde reduction [8]; 1-penten-3-ol was the characteristic marker of sample D; pentanal, n-nonanal, (E)-2-octenal, octanal, (E)-2-heptenal, 1-octen-3-ol, heptanal, 2-heptanone, (E)-2-hexenal, hexanal, (E)-2-pentenal, 3-methylbutanal, 2-methylbutanal, (Z)-3-hexen-1-ol, and 2-hexanone were the characteristic markers of sample E. No characteristic marker was detected for sample C in this heat map.

### 2.6. Analysis of Key Aroma Components of Walnut Oil from Different Varieties

The overall flavor of walnut oil was determined by the relative content of volatile components and its sensory threshold. Only certain compounds significantly contributed to the overall flavor of walnut oil; these compounds were key flavor substances of the sample. When measuring the contribution of volatile substances to flavor, it is not sufficiently comprehensive or accurate to judge only the relative content of substances. Standard research samples often contain dozens, or even hundreds, of volatile compounds; thus, performing the absolute quantification of the odor activity value is very time-consuming and expensive. Therefore, the ROAV was calculated to further analyze the contribution of volatile substances to the flavor of walnut oil. According to the sensory threshold obtained from the literature review [31], volatile substances with an ROAV greater than 0.1 in the volatile substance composition of walnut oil were identified. As shown in Table 2, hexanal and pentanal were the substances which contributed most to the flavor in the five different varieties of walnut oil, followed by heptanal. Elmore [25] studied the aroma volatiles of walnuts from China, Chile, and Ukraine using GC–MS analysis; the results showed that Ukrainian and Chinese nuts were similar quantitatively and qualitatively. The five compounds present at the highest levels in both nuts were the same. Hexanal was the largest compound in both samples, followed by 1-pentanol, pentanal, 1-hexanol, and then 1-penten-3-ol. Kalogiouri [32] explored the volatile metabolome of conventional and organic walnut oils through solid-phase microextraction and analysis with GC–MS combined with chemometrics; the results showed that hexanal was the most abundant volatile compound detected in conventional (0.212–0.403 μg kg^−1^) and organic walnut oil (2.923–5.340 μg kg^−1^). Zhou [33] carried out 10-day accelerated storage to investigate the evolution of the volatile profiles of walnut oils from three cultivars (cvs. Santai, Xiangling, and Qingxiang); the result showed that, in the fresh walnut oil, hexanal was the most abundant compound in oils from each cultivar. These conclusions are similar to our study, but there are also differences, which may be due to the different varieties, origins, and processing techniques.

In addition, Kruskal–Wallis tests were performed to gain further insight into the distribution of the key VOCs of samples A, B, C, D, and E. Then, we derived the box plot shown in Figure 5. The relative content differences of eight key VOCs in five varieties of walnut oil can be directly compared in Figure 5. Furthermore, the distribution of the relative contents of four substances, hexanal (M), hexanal (D), pentanal (M), (E)-2-hexanal (M), in the five varieties of walnut oil presented extremely significant differences (*p* < 0.01). Additionally, the distribution of the relative contents of pentanal (D), heptanal (M), and octanal (M) showed significant differences (0.01 ≤ *p* < 0.05).

## 3. Materials and Methods

### 3.1. Materials

Five varieties of walnut origin from two regions of Xinjiang were collected; the information from the samples is shown in Table 3. The sampled walnut trees were planted artificially. According to the requirements of walnut cultivation technology, farm manure and pesticide were applied in key phenological periods. All the walnuts were harvested in September 2021 and preserved at 4 °C to prevent deterioration before use after peeling and drying. The appearances of the five varieties of walnut are shown in Figure 6.

### 3.2. Preparation of Walnut Oil

Walnuts were shelled, and the walnut kernels were manually picked to remove the hollow, moldy, and abnormal walnut kernels. Virgin walnut oil was obtained using the hydraulic press method. Briefly, 30 kg of walnut kernels was pressed at 30 MPa for 45 min at room temperature. Then, the virgin walnut oil was centrifuged at 4 °C and 10,000 rpm for 15 min to obtain walnut oil samples for testing. Subsequently, a 5.00 mL sample was put into a 20 mL headspace vial. Additionally, three parallel analyses were performed for each sample.

### 3.3. HS–GC–IMS Analysis

The VOCs of walnut oil from five varieties were determined with HS–GC–IMS, using a FlavourSpec^®^ Gas Chromatograph Analyser (G.A.S., Dortmund, Germany) equipped with an automatic headspace sampling unit.

The GC-IMS conditions that were conducted refer to the method of Wang et al. [24] with minor modifications.

The GC was equipped with a MXT-5 column (15 m, 0.53 mm ID, 1.0 μm df). Samples were incubated at 80 °C for 15 min, with the rotary speed of 500 rpm. Then, 500 μL of headspace extracted sample was automatically injected via a heated syringe at 85 °C. Before each analysis, the syringe was automatically flushed for 0.5 min to avoid cross contamination. The carrier gas flow was programmed as follows: initial flow rate of 2 mL/min, which increased linearly to 10 mL/min within 8 min after 2 min, then increased linearly to 100 mL/min within 10 min, and finally increased linearly to 150 mL/min within 5 min. The analytes were separated in the column at 60 °C in 25 min. The carrier gas is high-purity (≥99.999%) nitrogen. It was then ionized in the IMS ionization chamber with a tritium gas source (6.5 KeV) in positive ion mode. The 10 cm drift tube was operated at a constant voltage (350 V/cm) at 45 °C with a nitrogen flow of 150 mL/min.

The identification of VOCs was performed by comparing the drift time with the GC-IMS library, and the relative content is determined by the proportion of the average relative peak area of each substance in the sum of the average peak area of the variety.

### 3.4. Determination of Key Flavor Compounds

The sensory threshold and relative content of volatile compounds jointly determine their action intensity in odor. The contribution of volatile substances to the flavor of the sample was determined by calculating the ROAV. It is generally believed that volatile substances with ROAV ≥ 1 are key flavor substances in the sample; 0.1 ≤ ROAV < 1 can modify the overall flavor of the sample; ROAV < 0.1 have limited effects on the flavor of the sample [34]. The ROAV of the component that contributed most to the flavor of the sample is defined as 100, and the ROAV of other components is calculated according to the following formula:(1)$$ROAV_{A}=\frac{C_{A}}{T_{A}}\times \frac{T_{S\tan}}{C_{S\tan}}\times 100$$
where *C_A_* is the relative content /% of each component, *T_A_* is the sensory threshold/(mg/kg) of each component, *C*_Stan_ is the relative content/% of the compound that contributes the most to the overall flavor, and *T*_Stan_ is the sensory threshold/(mg/kg) of the compound that contributes the most to the overall flavor.

### 3.5. Data Processing

Laboratory analytical viewer (LAV) analysis software equipped with an instrument, the NIST 2014 database, and the IMS database were used for the qualitative analysis of volatile compounds and to establish the VOC fingerprints. The Dynamic PCA plug-in was used for PCA processing, and the Matching matrix in LAV software was used for similarity analysis. A clustering heat map and box plot of the dataset were drawn using R studio (v1.2.13). Other charts were drawn using Excel 2010 and Origin 8.0.

## 4. Conclusions

In conclusion, 50 VOCs in walnut oil from five cultivars were detected, and 41 VOCs were specifically identified, including 14 aldehydes, 8 alcohols, 4 ketones, and 2 esters. The highest relative content among these VOCs was hexanal (relative contents: A, 34.36%; B, 33.40%; C, 42.30%; D, 43.30%; E, 38.52%)

Among the identified VOCs, the contents of twenty substances in sample E were the highest. According to PCA, sample E was quite different from the other four samples.

The characteristic VOC markers of the four samples were obtained using heat map analysis. Specifically, isopropyl acetate was the characteristic marker of sample A; acetone, 3-methylbutanol, and 1-hexanol were the characteristic markers of sample B; 1-penten-3-ol was the characteristic marker of sample D; fifteen substances, including pentanal and n-nonanal, were the characteristic markers of sample E; and there was no characteristic marker detected for sample C.

The key flavor substances of the sample were defined by calculating the ROAV. Hexanal and pentanal were the substances which contributed the most to flavor in walnut oil from five different varieties, followed by heptanal.

The relative content differences of eight key VOCs in walnut oil from five varieties could be directly compared using the box plot. The relative content distribution of hexanal (M), hexanal (D), pentanal (M), and (E)-2-hexanal (M) in walnut oil from five varieties represented a very pronounced statistically significant difference (*p* < 0.01). Additionally, the relative content distributions of pentanal (D), heptanal (M), octanal (M) showed significant differences (0.01 ≤ *p* < 0.05).

These results are conducive for the rapid detection of walnut oil quality, the identification of walnut food authenticity, and distinguishing the varieties of walnut oil.

## Figures and Tables

**Figure 1 molecules-28-02949-f001:**
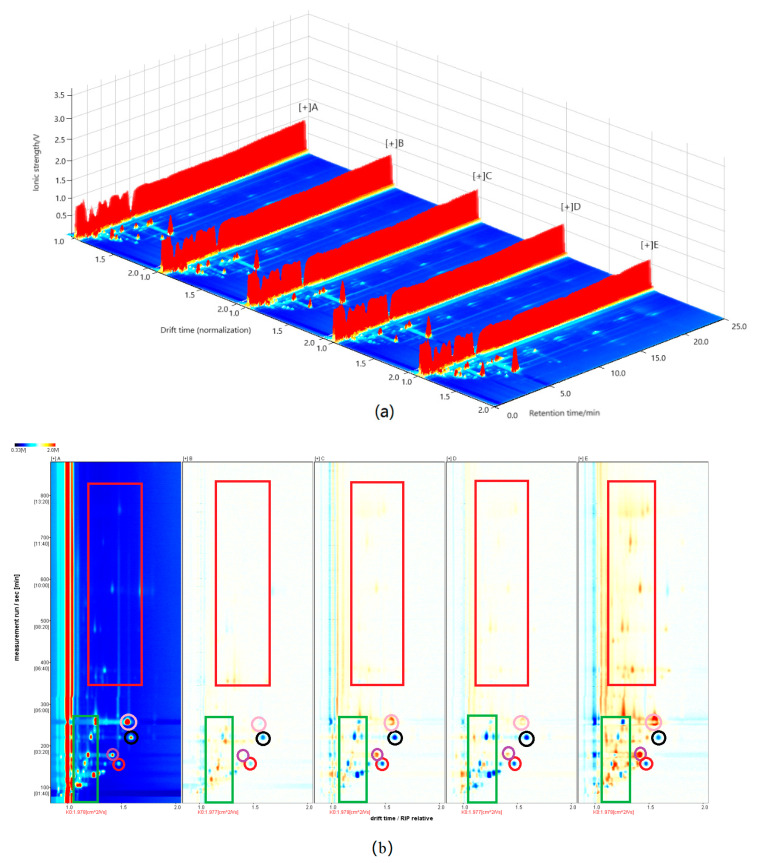
(**a**): GC–IMS three-dimensional spectra of the VOCs in walnut oil from five different varieties; (**b**): Comparison of GC-IMS spectra of VOCs in of walnut oil from five different varieties (after color deduction).

**Figure 2 molecules-28-02949-f002:**
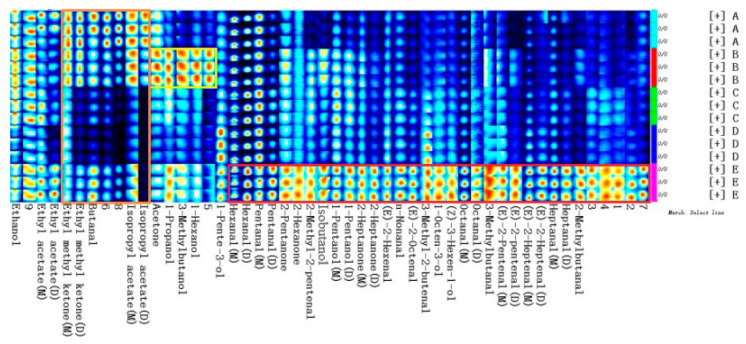
Fingerprint (gallery plot) of characteristic VOCs in walnut oil from five different varieties (A: Xin2; B: 185; C: Xinfeng; D: Xin’guang; E: Zha343).

**Figure 3 molecules-28-02949-f003:**
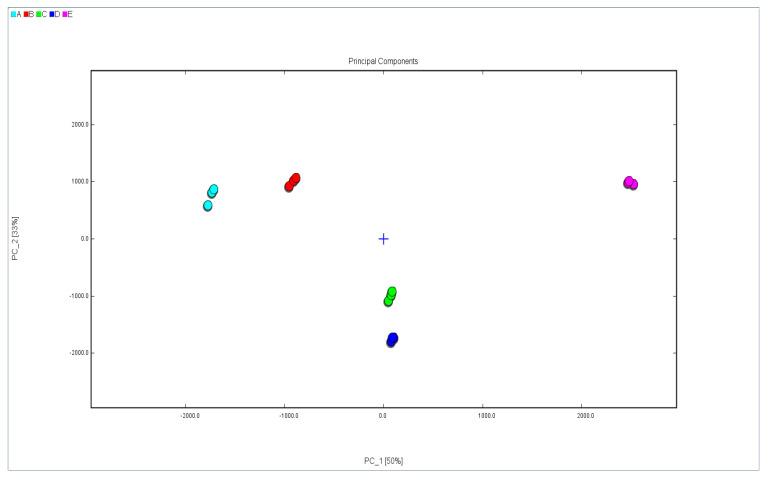
PCA diagram of walnut oil from five different varieties((A: Xin2; B: 185; C: Xinfeng; D: Xin’guang; E: Zha343)).

**Figure 4 molecules-28-02949-f004:**
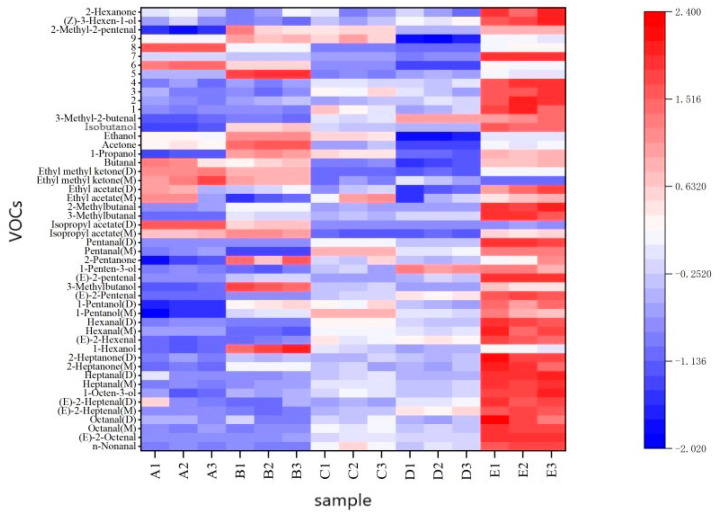
Heat map of VOCs in walnut oil from five different varieties.

**Figure 5 molecules-28-02949-f005:**
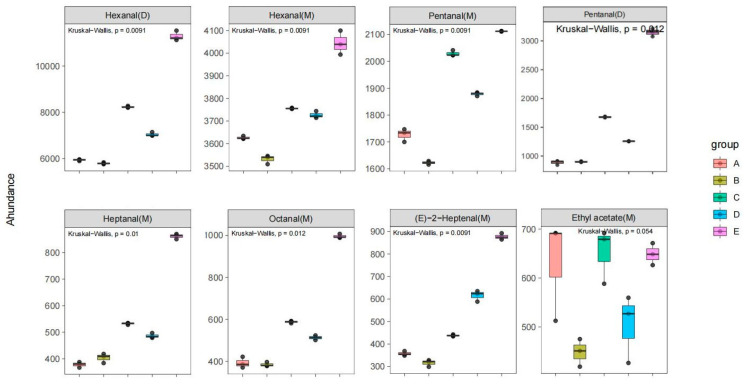
Box plot of key VOCs (relative content) in walnut oil from five different varieties.

**Figure 6 molecules-28-02949-f006:**
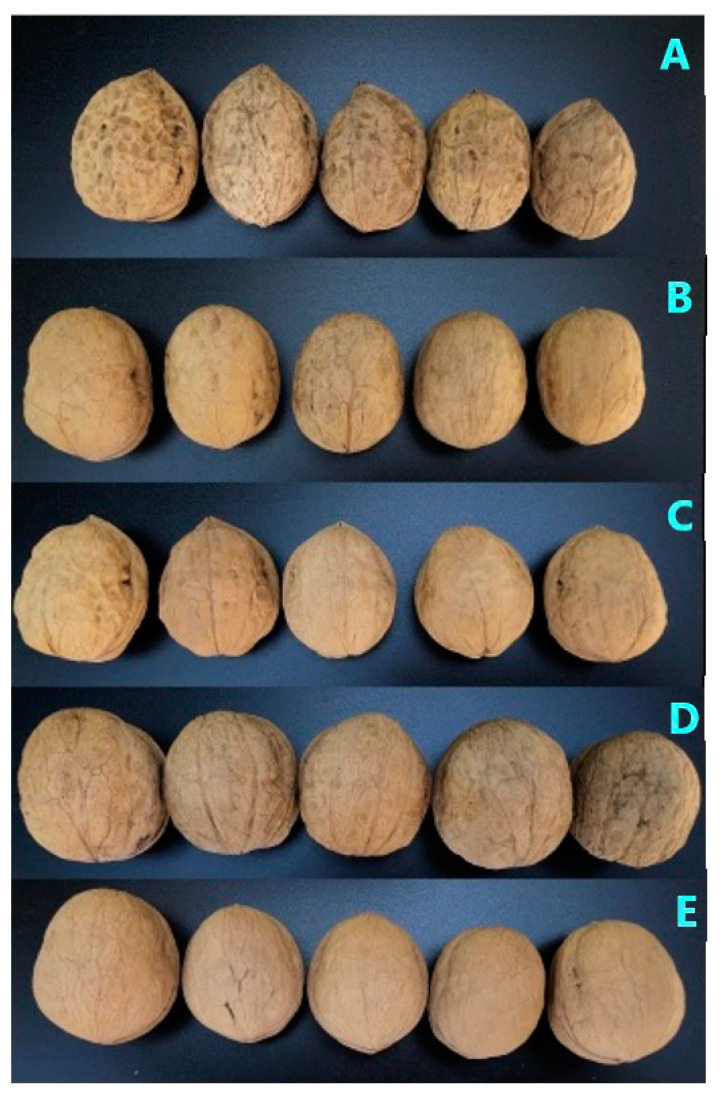
The appearances of walnuts from five varieties.

**Table 1 molecules-28-02949-t001:** Relative content of VOCs in walnut oil from five different varieties.

Number	Compounds	CAS#	Molecular Formula	Retention Index	Retention Time (s)	Relative Content (%)				
						A	B	C	D	E
1	n-Nonanal	C124196	C_9_H_18_O	1103.7	767.133	0.80	0.82	1.18	1.16	1.18
2	(E)-2-Octenal	C2548870	C_8_H_14_O	1070	690.61	0.51	0.54	0.84	0.87	1.25
3	Octanal(M)	C124130	C_8_H_16_O	1013	578.335	1.41	1.38	2.07	2.06	2.50
4	Octanal(D)	C124130	C_8_H_16_O	1011.6	575.826	0.19	0.18	0.22	0.22	0.26
5	(E)-2-Heptenal(M)	C18829555	C_7_H_12_O	962	481.114	1.28	1.13	1.55	2.48	2.20
6	(E)-2-Heptenal(D)	C18829555	C_7_H_12_O	960.6	478.605	0.26	0.18	0.22	0.37	0.45
7	1-Octen-3-ol	C3391864	C_8_H_16_O	993.2	542.583	0.37	0.45	0.52	0.55	0.61
8	Heptanal(M)	C111717	C_7_H_14_O	900.7	380.185	1.36	1.45	1.88	1.95	2.16
9	Heptanal(D)	C111717	C_7_H_14_O	901.3	381.061	0.13	0.10	0.17	0.16	0.30
10	2-Heptanone(M)	C110430	C_7_H_14_O	892.5	368.356	0.50	0.70	0.60	0.67	0.70
11	2-Heptanone(D)	C110430	C_7_H_14_O	891.2	366.603	0.09	0.12	0.12	0.14	0.23
12	1-Hexanol	C111273	C_6_H_14_O	881.4	353.897	0.22	0.53	0.31	0.32	0.24
13	(E)-2-Hexenal	C6728263	C_6_H_10_O	852.7	319.285	0.48	0.50	0.63	0.74	0.58
14	Hexanal(M)	C66251	C_6_H_12_O	794.1	258.823	13.03	12.65	13.25	14.98	10.16
15	Hexanal(D)	C66251	C_6_H_12_O	796	260.575	21.33	20.75	29.05	28.32	28.36
16	1-Pentanol(M)	C71410	C_5_H_12_O	771.2	237.626	1.52	2.21	2.58	2.51	1.88
17	1-Pentanol(D)	C71410	C_5_H_12_O	772.1	238.462	0.19	0.44	0.43	0.37	0.39
18	(E)-2-Pentenal	C1576870	C_5_H_8_O	741.5	211.989	0.49	0.58	0.72	1.01	0.90
19	3-Methylbutanol	C1576870	C_5_H_8_O	741.8	212.268	0.23	0.70	0.38	0.36	0.36
20	(E)-2-pentenal	C123513	C_5_H_12_O	739.1	210.038	0.05	0.13	0.08	0.12	0.30
21	1-Penten-3-ol	C616251	C_5_H_10_O	689	173.255	0.98	0.93	1.09	1.68	1.05
22	2-Pentanone	C107879	C_5_H_10_O	684.4	170.925	0.17	0.25	0.21	0.23	0.16
23	Pentanal(M)	C110623	C_5_H_10_O	697.2	178.808	6.20	5.81	7.16	7.55	5.31
24	Pentanal(D)	C110623	C_5_H_10_O	695.9	177.956	3.19	3.23	5.92	5.05	7.91
25	Isopropyl acetate(M)	C108214	C_5_H_10_O_2_	657.2	157.714	1.56	1.68	0.76	0.88	1.02
26	Isopropyl acetate(D)	C108214	C_5_H_10_O_2_	656.2	157.288	1.06	0.68	0.06	0.08	0.06
27	3-Methylbutanal	C590863	C_5_H_10_O	651.3	155.001	0.67	1.12	0.95	1.21	1.47
28	2-Methylbutanal	C96173	C_5_H_10_O	656.9	157.608	0.16	0.43	0.20	0.23	0.72
29	Ethyl acetate(M)	C141786	C_4_H_8_O_2_	613.8	138.799	2.27	1.61	2.30	2.03	1.63
30	Ethyl acetate(D)	C141786	C_4_H_8_O_2_	614.9	139.227	1.32	1.02	0.83	0.43	1.28
31	Ethyl methyl ketone(M)	C78933	C_4_H_8_O	597.2	132.173	4.88	4.61	3.46	4.33	2.39
32	Ethyl methyl ketone(D)	C78933	C_4_H_8_O	593.3	130.677	8.74	7.89	1.86	1.87	3.94
33	Butanal	C123728	C_4_H_8_O	602.1	134.097	2.81	2.35	1.12	0.78	1.83
34	1-Propanol	C71238	C_3_H_8_O	568	121.273	0.97	2.03	1.75	1.13	1.34
35	Acetone	C67641	C_3_H_6_O	515.7	103.954	2.87	3.55	2.32	2.07	1.89
36	Ethanol	C64175	C_2_H_6_O	486.7	95.438	5.53	6.37	5.69	4.09	3.72
37	Isobutanol	C78831	C_4_H_10_O	632	146.463	0.37	1.21	0.81	0.84	1.10
38	3-Methyl-2-butenal	C107868	C_5_H_8_O	755.4	223.599	0.45	0.50	0.72	1.11	0.73
39	1	unidentified	-	1099.5	757.097	0.23	0.22	0.38	0.31	0.41
40	2	unidentified	-	945.3	451.217	0.17	0.22	0.22	0.29	0.39
41	3	unidentified	-	905.5	387.195	0.19	0.18	0.29	0.26	0.30
42	4	unidentified	-	905.2	386.757	0.14	0.16	0.20	0.22	0.24
43	5	unidentified	-	882.7	355.65	0.18	0.47	0.12	0.17	0.17
44	6	unidentified	-	752.9	221.463	4.02	2.99	1.30	0.80	1.75
45	7	unidentified	-	752.9	221.463	0.74	0.71	0.56	0.54	1.16
46	8	unidentified	-	751.6	220.349	3.39	1.69	0.41	0.24	1.19
47	9	unidentified	-	517.2	104.419	1.99	2.14	2.08	1.75	1.39
48	2-Methyl-2-pentenal	C623369	C_6_H_10_O	827.2	291.404	0.08	0.22	0.20	0.16	0.16
49	(Z)-3-Hexen-1-ol	C928961	C_6_H_12_O	852.9	319.566	0.13	0.11	0.14	0.20	0.20
50	2-Hexanone	C591786	C_6_H_12_O	782	247.67	0.08	0.08	0.08	0.09	0.07

Notes: - Not detected.

**Table 2 molecules-28-02949-t002:** ROAV of VOCs in walnut oil from five different varieties.

Number	Compounds	Threshold value (μg/kg)	ROAV				
			A	B	C	D	E
1	Octanal (M)	0.7	2.17	2.18	2.34	2.39	2.90
2	(E)-2-Heptenal (M)	3	0.46	0.42	0.41	0.67	0.60
3	(E)-2-Heptenal (D)	3	0.09	0.06	0.06	0.10	0.12
4	Heptanal (M)	0.26	5.63	6.16	5.72	6.11	6.75
5	Hexanal (M)	0.23	61.08	60.95	45.61	52.89	35.81
6	Hexanal (D)	0.23	100.00	100.00	100.00	100.00	100.00
7	Pentanal (M)	0.85	7.87	7.58	6.67	7.21	5.06
8	Pentanal (D)	0.85	4.05	4.21	5.51	4.83	7.55
9	Ethyl acetate (M)	5	0.49	0.36	0.36	0.33	0.26
10	Ethyl acetate (D)	5	0.29	0.23	0.13	0.07	0.21

**Table 3 molecules-28-02949-t003:** Information of walnut from five varieties.

Identification	Varieties	Place of Origin	Harvest Time	Single Fruit Weight/g	Shape	Evaluation of Surface Texture
A	Xin2	Kashgar, Xinjiang	15 September 2021	15.90–18.40	Ovoid nut	Smooth shell surface, tight chink
B	185	Aksur, Xinjiang	22 September 2021	11.20–14.20	Round nut, has a spike	Smooth shell surface, loose chink
C	Xinfeng	Kashgar, Xinjiang	18 September 2021	12.60–14.90	Short oval nut, has a spike	Obvious surface gullies, tight chink
D	Xin’guang	Kashgar, Xinjiang	18 September 2021	16.80–17.50	Round nut	Smooth shell surface, tight chink
E	Zha343	Kashgar, Xinjiang	12 September 2021	12.40–15.30	Ovoid nut	Smooth shell surface, tight chink

## Data Availability

All the available data are incorporated in this manuscript.

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
