# Peer review of "Comparative Study on the Volatile Organic Compounds and Characteristic Flavor Fingerprints of Five Varieties of Walnut Oil in Northwest China Using Using Headspace Gas Chromatography-Ion Mobility Spectrometry"

_molecules, 2023, doi:10.3390/molecules28072949_

Round 1

Reviewer 1 Report (Previous Reviewer 2)

I have no additional comments

Author Response

We deeply appreciate the reviewers for the critical review of the manuscript. Accordingly, we have substantially revised our manuscript and marked the changes using “Track Changes” function in the revised revision. The revised manuscript has been checked by a native English-speaking colleague using an editing service. 

Reviewer 2 Report (New Reviewer)

Manuscript ID: molecules-2194102 Comparative Study on Volatile Organic Compounds and Characteristic Flavor Fingerprint of Walnut Oil from Five Varieties in Northwest China Using Headspace Gas Chromatography-ion Mobility Spectrometry

Sun et al. investigated the characteristic volatile organic compound (VOC) fingerprint of five varieties of walnuts grown in Northwest China using Headspace Gas Chromatography-ion Mobility Spectrometry. The authors reported statistically significant differences between sample varieties and the compounds responsible for sample discrimination were identified.

The manuscript is of interest, and it well fits the aims and scope of the journal.

Abstract needs improvement as it is not self-explanatory. The introduction refers to published literature within the field of research and the results are clearly reported. Overall, the paper is well written, the graphical representations are suitable for the purpose, and conclusions are consistent with the evidence presented. Cited references are relevant and within the last 5 years. Although some points need to be addressed.

Abstract: “In order to compare the differences of volatile flavor characteristics in different varieties (…)” – Why are you interested? Please set the background first.

Line 42: realize – not sure what the authors mean here. Misspelled.

Line 47: The study of the composition…

Table 1 and Figure 1 caption: English needs to be improved.

Line 71: “Put 5.00 mL (…)” – This needs to go into the past tense similarly to what was written beforehand.

Line 72: “Three repetitions were made for GC-IMS analysis.” – Repetitions or replicates?

Line 76 – 78: Again, do not mix up the tenses present and past.

Figure 3 and Figure 4: XY axis titles and labels are barely visible.

Figure 6 – Sample E is missing on the PCA representation.

Figure 7 – What are the axis? Detailed description on the figure caption is needed.

Line 295: “extremely significant difference…” Would sound better: a very pronounced statistically significant difference. 

Line 299: Since the results reported here are specific to varieties of walnuts grown in China, this can be used in food authenticity as well.

Relevant articles that should be cited accordingly:

Hao, J. et al., HS-SPME GC–MS characterization of volatiles in processed walnuts and their oxidative stability, Journal of Food Science and Technology, 57 (7) (2020), pp. 2693-2704

Xu, Y. et al., Comparison of aroma active compounds in cold- and hot-pressed walnut oil by comprehensive two-dimensional gas chromatography-olfactory-mass spectrometry and headspace gas chromatography-ion mobility spectrometry, Food Research International, 163, (2023), 112208

Author Response

Reviewer 3 Report (New Reviewer)

Round 2

Reviewer 3 Report (New Reviewer)

The authors made some changes in the articles, however, it is far to fit with the quality of the journal "Molecules". 

1. Introduction does not provide a sufficient survey of the literature. There is no proper justification for the choice of measurement method. It is mentioned "rapid detection" without providing any value. Infrared spectroscopy and e-nose are much faster than MS, however, sensitivity and specificity are lower than MS. Both techniques are developing fast. Therefore, a discussion on that is necessary. The citations used for the infrared technique for VOC detection are not appropriate. Both references are not related to gas phase measurements.

2. Description of the method is not informative enough to reproduce the results.

3. Results and discussions are still not clear enough. It only reflects the observation. This is not enough for the quality of "Molecules".

4. Quality of the figures is substandard and some cases not readable.

5. English needs to be improved.

Author Response

This manuscript is a resubmission of an earlier submission. The following is a list of the peer review reports and author responses from that submission.

Round 1

Reviewer 1 Report

The title of the paper is "Comparative Study on Volatile Organic Compounds and Characteristic Flavor Fingerprint of Walnut Oil from Five Varieties in Northwest China Using Headspace Gas Chromatography-ion Mobility Spectrometry"

I can't spot much scientific novelty in this paper. Even though IMS was used, no fresh contribution to the general knowledge steming from this fact can be seen.

Specific remarks:

1. "Walnut oil has the effects of cell repair, anti-inflammatory, anti-virus and antithrombotic." - please avoid such unsubstantiated and misleading statements. Such practice is unacceptable when it comes to scientific journal. You do not cite any medical research that proves your statements. Either rephrase this statement or cite clinical trial results.

2. "Headspace solid phase microextraction (HS-SPME), simultaneous distillation extraction and supercritical fluid extraction are the most commonly used, but their extraction efficiencies are limited. As a new rapid detection technology, gas chromatography-ion mobility spectrometry (GC-IMS) realizes rapid detection and analysis" - extraction efficiency and speed of detection (?) are two very different aspects of an analytical method. Mixing them up makes no sense.

3. "GC-IMS is still in the preliminary development stage at home and abroad" - the term "home and abroad" is not clear. I guess it means "in China and abroad" but the reader should not make gueses.

4.  Figure 2 - no credits are given here as if it was the work of the Authors. But, the same figure has been published in "Characterization of the volatile organic compounds produced from green coffee in different years by gas chromatography ion mobility spectrometry" (RSC Advances, 24, 2022).

5. "Samples were separated by a chromatographic column maintained at 60 °C. [...] During the 25-min chromatographic separation, the initial carrier gas flow rate was 2 mL/min for 2 min, and then increased linearly to 10 mL/min within 8 min, to 100.00 ml /min within 10 min, and finally to 150 mL/min within the final 5 min." - is it a correct description? Were samples separated isotermally with the gradient of a carrier gas flow? Please double-check this statement.

6. The formula (lines 99-101) is missing.

7. Remove or improve Figure 3, its current quality is too poor for drawing any meaningful conclusions.

8. "The differences of VOC in different varieties of walnut oil could be
seen intuitively from Fig.3." - intuitively, looking at Figure 3, all samples are the same.

9. "Fig.3 and Fig.4 showed that the VOCs of walnut oil samples can be well separated by GC-IMS technology. There were great differences in walnut oil characteristic flavor among samples from five different varieties according to GC-IMS spectral information shown in the Fig.3 and Fig.4." - unfortunately I can't see any "great differences". As mentioned before (p. 8) all the samples/chromatograms look alike. Only the minor differences can be spotted. In addition chromatogram for sample E is missing.

10. "Fig. 5 revealed that there are obvious differences in VOCs of walnut oil from five different varieties, among which A and B were relatively similar,
and C and D were relatively similar." - please decide whether there are "great differences" or similarities between samples.

11. There's no information on scaling used to produce Figure 5, i.e. the reader doesn't know the magnitude of the differences shown in this figure. For example, does the blue spot represent difference of 10, 50 or maybe 100%? This makes the figure irrelevant.

12. Figure 6 - what is the point in providing results for three replicates for each sample in this figure? Why don't you averaged them? Single measurement is no measurement.

13. Table 2 is hardly readable please improve. What is Rt/s?

14. Overuse of the word "intuitively" becomes annoying.

15. How the relative amounts of the aroma compounds were calculated? No standards were used, haven't they?

16. "3.4. Heat map analysis" contains exactly the same information as the "3.2. Comparison of VOCs fingerprints in walnut oil from five different varieties" making one of this sections redundant and unnecessary.

17. "According to the sensory threshold obtained from literature review[17], volatile substances with ROAV greater than 0.1 in the volatile substance composition of walnut oil were calculated, and the results were shown in Table 3." - compounds 13 through 18 have ROAV values of 0 (zero) why they are listed in the table?

18. "Obviously, pentanal was the most flavor contributing substance in walnut oil from five different varieties, followed by heptanal." - that is confusing. Penatanal has ROAV value of around 7, while hexanal has the ROAV value of 100. So why the conclusion that pentanal is the most flavor contributing substance?

19. What kind of data was fed to PCA and NNA algorithms?

Author Response

Dear reviewer:

Thank you for your decision and constructive comments on my manuscript. We have carefully considered the suggestion of Reviewer and make some changes. We have tried our best to improve and made some changes in the manuscript.

The red part that has been revised according to your comments. Revision notes, point-to-point, are given in the document attached.

Reviewer 2 Report

The volatile constituents of oils of five walnuts varieties from different regions in China were studied by headspace gas chromatography and ion mobility spectrometry. The results showed that all oils have the same compounds, so, there were no qualitative differences between them. Instead, quantitative differences in some of the components presented the major varietal influences and could be useful to determine the identity of walnut genotypes. Unfortunately, the ROAV analysis to determinate the key flavor compounds in walnut oils was not present in a clear way, therefore, I did not review the results showed in Table 2 because I did not understand how the calculations were done. Otherwise, figures are cropped and of poor quality. Another weakness in the manuscript is that the results were only described but not discussed. Thus, authors are not considering relevant references that would be appropriate to comment. The conclusion could refer the advantages of the method used for the analysis of volatiles.

Below are some comments and suggestions regarding to the manuscript.

1.      Line 13, What is the criterion to say that there are obvious differences? sample A is similar to B, and C is similar to D. These results are not obvious to me. Perhaps replace “there were obvious differences” with “Quantitative differences were observed in”.

2.      Line 16. Please, include examples of aldehydes.

3.      Lines18-19. Again. What is the criterion to say that there are obvious differences?.

4.      Lines 19-21. Is there evidence of consumer preference towards any specific variety?. According to the literature, what compounds are directly related to the quality of a nut?. This information is missing in the document for this statement.

5.      Lines 21-22. Methodological evidence for this statement is lacking.

6.      Lines 38-39. This statement require references. Why do you think that their extraction efficiencies are limited?

7.      Lines 46-47. This statement require references.

8.      A table of some biometrical characteristics of walnuts would be appropriate.

9.      Fig 1. Is the GC-IMS scheme original?. Please include a reference.

10. Line 88. Please include the value of V/cm.

11.  Please, write the formula and check that references 10 and 16 correspond to the ROAV formula.

12.  The compounds were tentatively identified with the best match of mass spectra from databases. Retention index values were not calculated and volatile compounds were not confirmed by using standard chemicals.

13.  Please indicate the coordinate axes in Fig. 3 (retention time, migration time, and peak intensity). Besides, 3-D topographic image corresponds to volatile compounds from nut oils but the image file is incomplete.

14.  The spectrum of sample E in Fig 4 is missing. Figs 4 and 5 indicate the same information. I recommend to choose one and combining it with Fig. 3. Thus, Fig 3 will have 2 panels, panel (A) 3-D topographic in different samples and (B) Topographic plot of GC-MS spectra. The font size in the images is very small.

15.  Lines 130-131. Color contrast cannot be used to establish a reliable comparison between samples. The authors do not show a criterion for choosing sample A as the "reference".

16.  Line 134, perhaps replace “obvious” with “clear”. 

17.  Lines 135 and 136. For this statement a statistical analysis is required.

18.  Line 150, what do you mean by “intuitively”?.

19.  Lines 152-155. Please check that compound names in the text correspond to those mentioned in Table 2. For example, isobutanol is 2-methyl-1-propanol. In addition, the names of the compounds in the text must be cited with the first letter in lowercase and in the table with capital letters (lines 156, 161, 243).

20.  Line 156, …glutaraldehyde, and hexanal.

21.  Line 156, please check the IUPAC name of the compounds. For example replace 1-pentene-3-ol, with 1-penten-3-ol, heptanaldehyde with heptanal.

22.  Lines 161-162. Please check the end of the sentence…compound 9 et al. What does it mean?

23.  Table 2. Please include retention drift. Relative content/%, what does it mean? It’s a typo?. Are you referring to (%).

24.  Fig. 7. The figure appears incomplete. The labels of the samples to which the columns belong are not shown, nor the legend of the color intensities.

25.  Lines 200-202. Why do you say that pentanal is obviously the most flavor contributing substance in walnut oil?. For me there is nothing obvious. The ROAV for hexanal is the highest and the calculation of ROAV was not included in the manuscript.

26.  Fig. 8. This is an incorrect interpretation. Component 1 has no direct correlation with the distribution of the experimental units. The variation on the horizontal axis explains the differences observed mainly between A-B-C-E, while on the vertical axis the differences between groups (A-B-E) and (C-D) stand out. There seems to be affinity between A-B, C-D and E is separated from the rest, however this ordering analysis is NOT a hypothesis test.

27.  Lines 222-223. PC2 cannot be "correlated" with any experimental unit. It is a wrong interpretation.

28.  Please replace this plot with a cladogram or a Venn diagram. What is the gray gradient bar at the top?.

29.  Please the figure captions need to be improved and more descriptive.

Author Response

(The authors gave the same response as above.)

Round 2

Reviewer 1 Report

This work is not scientifically sound and doesn't provide reliable data due to methodological faults which can not be fixed.

This is obvious when one look at the answers to the Comment no 5. Authors failed to answer the question about programming carrier gas flow – and this is the fundamental question when it comes to gas chromatography.

The optimal gas flow for the column they used is about 0.9–1.3 ml/min, but authors used gas flows ranging from 2 to 150 ml/min and they were unable to justify it.

This also explains the fact that no separation can be seen after around fifth minute (Figure 3). After this time, the carrier gas flow was so high that the column efficiency was not existent.

A little deeper search reveals that Chen, T., Qi, X., Lu, D. et al. in their article "Gas chromatography-ion mobility spectrometric classification of vegetable oils based on digital image processing." Food Measure 13, 1973–1979 (2019). https://doi.org/10.1007/s11694-019-00116-5 used fairly similar conditions to investigate vegetable oils, but in their case a completely different (and appropriate)  chromatographic column was used.

Since the fundamental data were collected in an improper way the rest of the article is of little value.

Reviewer 2 Report

The manuscript definitively was improved by the authors. However, the english language and style will require extensive english editing, mainly the abstract and the new information showed in red. Please, review the wording of lines 39-41 because it is not understood. The red letters in Fig 1, would look better in a color other than red. The names of the compounds cited in the conclusion must be in lowercase.